# Local Competition and Enhanced Defense: How *Metarhizium brunneum* Inhibits *Verticillium longisporum* in Oilseed Rape Plants

**DOI:** 10.3390/jof9080796

**Published:** 2023-07-28

**Authors:** Catalina Posada-Vergara, Stefan Vidal, Michael Rostás

**Affiliations:** Agricultural Entomology, Department of Crop Sciences, University of Goettingen, Grisebachstr 6, 37077 Goettingen, Germany; svidal@gwdg.de

**Keywords:** *Metarhizium brunneum*, endophytic entomopathogenic fungi, *Verticillium longisporum*, split-root, *Brassica napus*, Brassicaceae

## Abstract

*Metarhizium brunneum* is a soil-borne fungal entomopathogen that can be associated with plant roots. Previous studies have demonstrated that root colonization by beneficial fungi can directly affect soil-borne pathogens through competition and antibiosis and can activate a systemic response in plants, resulting in a primed state for a faster and/or stronger response to stressors. However, the mechanisms by which *Metarhizium* inoculation ameliorates symptoms caused by plant pathogens are not well known. This study evaluated the ability of *M. brunneum* to protect oilseed rape (*Brassica napus* L.) plants against the soil-borne pathogen *Verticillium longisporum* and investigated whether the observed effects are a result of direct interaction and/or plant-mediated effects. In vitro and greenhouse experiments were conducted to measure fungal colonization of the rhizosphere and plant tissues, and targeted gene expression analysis was used to evaluate the plant response. The results show that *M. brunneum* delayed pathogen colonization of plant root tissues, resulting in decreased disease symptoms. Direct competition and antibiosis were found to be part of the mechanisms, as *M. brunneum* growth was stimulated by the pathogen and inhibited the in vitro growth of *V. longisporum*. Additionally, *M. brunneum* changed the plant response to the pathogen by locally activating key defense hormones in the salicylic acid (SA) and abscisic acid (ABA) pathways. Using a split-root setup, it was demonstrated that there is a plant-mediated effect, as improved plant growth and decreased disease symptoms were observed when *M. brunneum* was in the systemic compartment. Moreover, a stronger systemic induction of the gene *PR1* suggested a priming effect, involving the SA pathway. Overall, this study sheds light on the mechanisms underlying the protective effects of *M. brunneum* against soil-borne pathogens in oilseed rape plants, highlighting the potential of this fungal entomopathogen as a biocontrol agent in sustainable agriculture.

## 1. Introduction

Roots grow in the soil and share the same substrate with numerous microbes. It has been estimated that a gram of soil can contain up to 10 billion microorganisms [1]. Moreover, plant roots deposit up to 40% of the carbon they fix during photosynthesis into the soil. This process, called rhizodeposition, creates a microenvironment that promotes a 10- to 100-fold increase in microbial density around plant roots and a microbial community composition that is distinct from that found in bulk soil [2,3]. This microenvironment, known as rhizosphere, is the site for multiple interactions between soil microbes and plant roots. Soil microorganisms can have beneficial effects on plants as they can facilitate nutrient uptake or protect against abiotic and biotic stresses [4,5].

Several *Metarhizium* species, primarily known as fungal pathogens of insects, are also rhizosphere competent. They exhibit enhanced growth around roots [6,7] and can colonize plant tissues [7,8,9]. Together with other genera of entomopathogens such as *Beauveria* and *Lecanicillium*, these fungi have been referred to as “endophytic entomopathogenic fungi” (EEF). Several studies have reported positive effects of EEF on plants, including ameliorating salt stress [10,11], promoting growth [12,13,14], improving nutrient acquisition [15,16,17] and reducing insect and pathogen damage [18,19].

In the past decade, an increasing number of studies have demonstrated that different *Metarhizium* species can inhibit several plant pathogens in vitro [20,21,22]. Moreover, colonization by *Metarhizium* can reduce pathogen-induced symptoms in various plants [20,21,22,23,24]. However, the mechanisms by which *Metarhizium* species antagonize plant pathogens are not well understood [25,26]. Plant protection can result from a combination of competition and antibiosis [20,21,22]. *Metarhizium* has been shown to strongly inhibit the growth of pathogens in vitro through the production of secondary metabolites that are toxic to microorganisms and insects [27]. Mycoparasitism has been suggested as another potential mechanism, although this has not been observed with *Metarhizium* [28]. Another mechanism suggested is the activation of plant defenses. Beneficial microorganisms can trigger induced systemic resistance (ISR) in plants [4], which activates plant defenses and helps protect against pathogens. This has been observed with obligate symbionts such as mycorrhiza [29,30] and facultative endophytes such as *Trichoderma* or other non-pathogenic fungal strains [4,31].

Some studies have shown that EEF may modify plant defense pathways. For instance, peanut plants inoculated with *M. anisopliae* strain M202-1 showed downregulation of genes involved in the hypersensitive response and synthesis of resistance proteins [32]. Increased defense responses have also been observed. The jasmonic acid (JA) and salicylic acid (SA) pathways are essential in the defense response and were activated by *B. bassiana* BG11 in *Arabidopsis thaliana* [33] and by the strains Bb0062 and Bb02 in *Nicotiana benthamiana* [34]. The inoculation of *M. brunneum* Mb7 induced the expression of pathogen resistance genes such as *PR1* that play a role in the SA pathway [24]. Levels of SA and JA hormones were also higher in oilseed rape plants inoculated with *M. brunneum* F52 [35] and in maize plants inoculated with *M. anisopliae* A1080 [36] and *M. robertsii* strain ARSEF 14325 [12].

Recent publications suggest that *Metarhizium* primes plant defense responses against insects [37,38,39]. The phenomenon of priming is associated with ISR, whereby plants exhibit a faster and/or stronger defense response when encountering a stressor if they have been previously exposed to a beneficial agent. In addition, it has been demonstrated that pre-inoculation with *M. brunneum* Mb7 increased reactive oxygen species production in tomato plants elicited with the fungal protein ethylene-inducing xylanase, which is an inducer of plant defense responses [24], indicating that *Metarhizium* could prime plants against pathogens. However, to our knowledge, there is currently little knowledge on how *Metarhizium* modifies plant responses towards a pathogenic fungus.

*Verticillium longisporum* is a soil-borne pathogen that causes Verticillium stem striping in oilseed rape (*Brassica napus* L. spp. *oleifera*) [40]. Melanized microsclerotia present in the soil germinate upon induction by plant root exudates, and the resulting hyphae grow towards the root, colonizing the surface of the root hairs and penetrating the roots through rhizodermal cells [41]. Subsequently, the hyphae grow towards the central cylinder and enter the xylem vessels, spreading into aboveground organs. Upon plant tissue death, the pathogen produces microsclerotia that can remain in plant debris in the soil for more than 10 years [40].

In this study, we investigated the interactions between *M. brunneum* and *V. longisporum* during their colonization of oilseed rape plants. Our hypothesis was that, as soil and root inhabitants, both fungi could directly affect each other’s growth and plant colonization. We also hypothesized that *M. brunneum*, as a plant symbiont, could modify the plant’s response to *V. longisporum.* To study the direct interaction, we carried out in vitro dual confrontation assays and co-inoculation of oilseed rape roots with both fungi, where we measured fungal colonization of the hypocotyl and plant disease progress. In order to distinguish direct effects from plant-mediated effects, we used a split-root experiment to further explore how each fungus affected the other’s colonization of the rhizosphere, roots, and hypocotyl. We also measured the local and systemic plant responses to each fungus by analyzing the expression of marker genes involved in plant defenses and explored whether *M. brunneum* could modify the plant’s response to *V. longisporum*.

## 2. Materials and Methods

### 2.1. Study System

*Metarhizium brunneum* Cb15III was obtained from the in-house collection of the Division of Agricultural Entomology. The pathogen *Verticillium longisporum* Vl43 was provided by the Division of Plant Pathology and Crop Protection, University of Goettingen. To obtain spore suspensions, fungi were grown on potato dextrose agar (PDA) (Carl Roth GmbH, Karlsruhe, Germany) at 23 °C for 14 days. Spores were removed from hyphae by scraping the surface of the colony with a sterile glass slide and were suspended in 20 mL of 0.1% Tween 80 (Carl Roth GmbH, Karlsruhe, Germany). The suspension was filtered through a plastic gauze and adjusted using 0.1% Tween 80 to a final concentration of 1 × 10^7^ spores mL^−1^ for *M. brunneum* and 1.5 × 10^6^ spores mL^−1^ for *V. longisporum*. Spore viability was assessed before each experiment by plating 100 μL of a 1 × 10^3^ spores ml^−1^ distributed in 10 μL drops on PDA plates and counting the germinating colonies 48 h later.

Winter oilseed rape (*Brassica napus* var. Falcon) (Norddeutsche Pflanzenzucht Hans-Georg Lembke KG, NPZ, Hohenlieth, Germany) susceptible to *V. longisporum* [42] was used in this study. For all experiments, plants were grown in a non-sterile soil mix consisting of a mixture of commercial soil (Fruhstorfer Erde Typ 25, Hawita Gruppe GmbH, Vechta, Germany), heat-treated (steamed) compost and sand (2:1:1, *v*:*v*). Plants were kept in a greenhouse with 18–25 °C, supplemented with light to obtain a 16:8 (L:D) h photoperiod. 

### 2.2. In Vitro Confrontation Assay

Antibiosis between *M. brunneum* and *V. longisporum* was evaluated with a confrontation assay. Single colonies of each fungal species were produced through streaking a drop of spore suspension from a glycerol stock over PDA media on a 90 mm plate. The plates were then incubated at 23 °C in the dark for 48 h. Single colonies were then transferred to a 90 mm PDA Petri dish. Each fungus was grown either alone or in dual culture with a distance of 4 cm between colonies. Plates were placed completely randomized inside the incubator. The plates were scanned at 14, 17 and 21 days after starting the confrontation assay, and the scanned images were used to measure fungal colony area with the ImageJ software (Ver. 1.53f51) [43]. Growth inhibition (GI) was calculated using the area of the colonies growing alone (control) and in dual culture with the formula:$$\;\mathrm{GI}=\frac{\left(area\;control-area\;in\;dual\;culture\right)}{area\;control}$$

For each treatment (single culture − dual culture × 2 fungi), 5 replicates (plates) were carried out. 

### 2.3. In Planta Co-Inoculation Assay

To evaluate the effect of *M. brunneum* on the disease progress of *V. longisporum* infecting oilseed rape plants, a greenhouse experiment was conducted. Oilseed rape seeds were surface-sterilized in 1% sodium hypochlorite for 2 min, followed by 75% ethanol for 2 min, and rinsed three times with sterile water. Seeds were then germinated in sterile silica sand and grown for 24 days. Afterwards, seedlings were uprooted, and the root tips were cut off. The roots were then immersed in fungal spore suspension for 30 min. The treatments included the following.

Control: roots mock-inoculated with 0.1% Tween 80.Mb: roots inoculated with *M. brunneum* Cb15III.Vl: roots inoculated with *V. longisporum* VL 43.Mb/Vl: roots inoculated with a mix of both fungi.

After inoculation, the seedlings were transplanted into square pots (11 × 11 cm, 0.5 L). Each treatment had 25 biological replicates, and each replicate consisted of an individual pot with a plant. The plants were arranged in a completely randomized design in a greenhouse cabin. Disease assessment, plant parameters and fungal quantification in plant hypocotyl were conducted as described below.

### 2.4. Split-Root Assay

A split-root system was used to differentiate effects caused by direct fungal interaction from indirect effects via induced changes in plant metabolism. Split-root ready seedlings were produced as in [37]. The inoculation of fungi was sequential to give *M. brunneum* time to associate with the plant roots before the inoculation with *V. longisporum.* Seedlings were transplanted into two bound square pots (11 cm) filled with non-sterile soil mix. One of the split-root pots was inoculated by drenching the roots with 3 mL of either *M. brunneum* spore suspension or 0.1% Tween 80 (mock inoculation). Seven days later, *V. longisporum* was inoculated by pipetting into the soil 3 mL of spore suspension. The experimental setup consisted of 5 treatments, with each plant having two root compartments (C1 and C2):Control: C1 = mock inoculation, C2 = untreated soil.Mb: C1 = *M. brunneum* (Mb-L), C2 = mock inoculation (Mb-S).VL: C1 = *V. longisporum* (Vl-L), C2 = mock inoculation (Vl-S).Local: C1 = both fungi present (Mb-L/Vl-L) C2 = both fungi absent (Mb-S/Vl-S).Systemic: C1 = *M. brunneum* present, *V. longisporum* absent (Mb-L/Vl-S), C2 = *V. longisporum* present, *M. brunneum* absent (VL-L/Mb-S).

“L” means local inoculation, and “S” means systemic or that the inoculation was performed in the adjacent split compartment. Each treatment had 35 plants, and plants from all treatments were randomly distributed in the greenhouse cabin.

To gather samples for gene expression analysis, a second split-root experiment was conducted. The pathogen was inoculated by root dipping to ensure better exposure to fungal spores; for this, seven days after *M. brunneum* inoculation, the roots of the compartment to be inoculated were carefully taken out and placed inside a 2 mL Eppendorf tube filled with *V. longisporum* spore suspension for 20 min, after which the roots were covered again with soil. Roots in one compartment of the control treatment had the same manipulation, dipped in 0.1% Tween 80.

### 2.5. Disease Assessment

In all experiments, plant height and disease severity were evaluated weekly until 35 days after fungal inoculation (co-inoculation) or *V. longisporum* inoculation (split-root), using the scale described in [41]: 1 = no symptoms.2 = slight symptoms on oldest leaves (yellowing, black veins).3 = slight symptoms on next younger leaves.4 = about 50% of leaves showing symptoms.5 = more than 50% of leaves showing symptoms.6 = up to 50% of leaves dead.7 = more than 50% of leaves dead.8 = only apical meristem still alive.9 = plant dead.

Disease severity values were used to calculate the AUDCP values (area under the disease progress curve) according to the following formula [44]:AUDPC = Σ((y_i_ + y_i+1_)/2) × (*t*_i+1_ − *t*_i_)
where y_i_ is the value of the disease severity for the “i” observation, and *t*_i_ is the time after inoculation when the observation “i” was taken.

The dry weight of roots (co-inoculation), shoots (all) and leaf area (co-inoculation) were determined at 21, 28 and 35 days post-inoculation (dpi).

### 2.6. Quantification of Rhizospheric and Endophytic Metarhizium brunneum

In the split-root experiment, we evaluated whether *M. brunneum* colonized the rhizosphere and if it were affected by *V. longisporum.* We measured fungal colony forming units (CFUs) in the soil closely attached to the roots at 7, 21 and 35 dpi of *V. longisporum*. The complete root system with soil attached was placed in 50 mL Falcon tubes (Sarstedt AG & Co. KG, Nümbrecht, Germany) with 25 mL of 0.1% Tween^®^ 80. In order to release the rhizospheric soil from the roots, tubes were vortexed for 10 s, after which the roots were taken out for DNA extraction (see below). Tubes with rhizospheric soil were placed in a shaker in horizontal position for 20 min at 250 rpm. Samples were sonicated for 30 s, briefly vortexed and left to sediment for 20 s. Then, 100 μL of the supernatant was diluted 1:10 (*v*:*v*), and 100 μL from the dilution was plated on 9 cm Petri dishes with semi-selective medium [45]. The Petri dishes were incubated at 23 °C and 65% RH for 21 d in darkness. Fungal colonies were counted in three-day intervals, starting 10 days after plating until no new ones appeared. Colonies were confirmed as *M. brunneum* according to their morphology. Tubes with the original soil dilution were dried in an oven (at 60 °C for 5 days), until they reached constant weight, and the dry weight was recorded.

Fungal DNA in root (split-root experiment) and in hypocotyl (first pot experiment and split-root experiment) was measured with real-time quantitative polymerase chain reaction (qPCR). Roots were washed, dried and placed in −20 °C. For each plant, a 2 cm segment of hypocotyl was sliced, surface-sterilized in 70% ethanol for 1 min and 2% sodium hypochlorite for 5 min, rinsed three times with sterile water for 30 s and placed in −20 °C. Roots and hypocotyls were lyophilized for 72 h (Martin Christ Freeze Dryers, Osterode am Harz, Germany) and milled with a mixer mill (Retsch MM 400, Haan, Germany) in a stainless steel container with a 20 mm, 32 g steel sphere for 30 s at maximum speed. DNA was extracted from 30 mg of root tissue with the cetyltrimethylammonium bromide (CTAB) buffer extraction method described previously [46], and DNA quality was verified in agarose (0.8%) gels. The CFX384™ Real-Time System with a C1000™ Thermal Cycler (BioRad, Hercules, CA, USA) was used for fungal DNA amplification and melting curve analysis. The primers used for *M. brunneum* detection are specific for the *Metarhizium* PARB clade: Ma 1763 (CCAACTCCCAACCCC TGTGAAT) and Ma 2097 (AAAACCAGCCTCGCCGAT) [47]. Specific primers used for *V. longisporum* were OLG 70 (CAGCGAAACGCGATATGTAG) and OLG 71 (GGCTTGTAGGGGGTTTAGA) [41]. Amplification was performed with 1:10 dilutions of the DNA extracts. The reaction mixture contained the following: 5 μL of 2× qPCRBIO SyGreen Low-ROX (PCR Biosystems, London, UK), 0.2 μL of 10 μM of each primer and 3.6 μL of 1 μL of DNA template solution, completed to a total of 10 μL of final volume reaction. qPCR running conditions started with an initial denaturation for 2 min at 95 °C, followed by 40 reaction cycles consisting of 5 s denaturation step at 95 °C, 20 s annealing step at 66 °C (for *M. brunneum*) or 60 °C (for *V. longisporum*) and 10 s extension at 72 °C. The final elongation was performed for 5 min at 72 °C. Melting curves were obtained by heating the samples to 95 °C for 60 s and cooling them to 55 °C for 60 s followed by a temperature increase from 55 °C to 95 °C by 0.5 °C per cycle with continuous fluorescence measurement. Absolute fungal DNA amount per g of plant tissue was measured by comparing threshold cycle (Ct) values against DNA standards starting with a concentration of 100 pg × μL^−1^ and decreasing with a 1:3 dilution factor. The threshold cycle and standard curves were generated by the Bio-Rad CFX Maestro software (Ver. 1.1). The identity of the amplicon was verified by comparing its size with gel electrophoresis. The presence of DNA in the root was evaluated in 5 replicates (first pot experiment) or in each root compartments from 8 plant replicates (split-root experiment) for each sampling date. 

### 2.7. Gene Expression in Plant Roots

Gene expression was analyzed in root tissues obtained from the second split-root experiment. Plants were harvested 7 days after *V. longisporum* inoculation as previous studies have found clear plant responses to the pathogen [48,49]. Whole roots were washed and snap-frozen in liquid nitrogen, lyophilized, placed in a 2 mL Eppendorf tube with 4 stainless steel 0.5 mm spheres and milled with a mixer mill for 60 s at 30 Hz. 

Total RNA was extracted with RNAzol ^®^RT (Sigma-Aldrich, St. Louis, MO, USA) from 20–30 μg of lyophilized ground plant tissue, following the manufacturer’s instructions. The integrity of RNA was evaluated by denaturing gel electrophoresis. Concentration and purity were assessed by measuring OD_260_/OD_230_ and OD_260_/OD_280_ absorbance ratios using a microplate spectrophotometer (Epoch, Bio-Tek, Winooski, VT, USA). First-strand cDNA was synthetized from 1 μg of total RNA using Fast Gene^®^ Scriptase II (Nippon Genetics Europe, Düren, Germany) and oligo dT, following the manufacturer’s instructions. Primers used were published in previous studies (Appendix A) with the exception of ICS2, which was designed using Primer3 v.4.1.0) [50] with *B. napus* specific gene sequences from the Genbank (https://www.ncbi.nlm.nih.gov/genbank/ (accessed on 21 November 2021)) data base.

Gene expression was measured by qPCR with the equipment described in 2.6. The reaction mixture contained the equivalent of 5 ng total RNA, 5 μL of 2× qPCRBIO SyGreen Low-ROX (PCRBIOSYSTEMS, London, UK), 0.2 μL of 10 μM of each primer and 3.6 μL of 1 μL of DNA template solution, completed to a total of 10 μL of final volume. The program consisted of 95 °C for 2 min, 40 cycles of 95 °C for 10 s and 60 °C for 30 s. Amplicon specificity was controlled by melting curve analysis as previously described. The relative expression of each gene was calculated using the 2^−∆∆CT^ method, with correction for primer efficiency, tested using a five-dilution series of the template [51], normalized to the endogenous reference gene *ACTIN* and subsequently normalized to those in the control plants. For this study, we selected the following genes in the hormonal and glucosinolate (GSL) pathway: abscisic acid (ABA) biosynthesis: *ABA2* (*xanthoxin dehydrogenase*); ethylene (ET) biosynthesis: *ACO* (*1-aminocyclopropane-1-carboxylic acid oxidase*); ET downstream signaling: *ERF2* (*ethylene-responsive transcription factor 2*); salicylic acid (SA) synthesis: *PAL* (*phenylalanine ammonia-lyase*) and *ICS2* (*isochorismate synthase 2*); SA downstream signaling: *PR1* (*pathogenesis-related protein 1*); jasmonic acid (JA) synthesis: *AOS* (*allene oxide synthase*); JA signaling: *PDF 1.2* (*defensin-like protein 16*); aliphatic GSL biosynthesis: *CYP83A1* (*cytochrome P450 83A1*) and indole GSLs biosynthesis: *CYP79B2* (*cytochrome P450 79B2*).

### 2.8. Data Analysis

Data exploration and statistical analyses were performed with the software R 4.0.3 [52]. DNA, CFU and gene expression data were log transformed, and plant weight was transformed to the power of two to meet the assumptions of normality and homogeneity of variance and analyzed with one-way ANOVA. Significant differences between treatments were evaluated post hoc with the Fisher LSD test (Library agricolae, [53]). AUDPC values were analyzed with a generalized linear model (GLM) with the quasi-Poisson family due to overdispersion.

## 3. Results

### 3.1. Direct and Systemic Effects of M. brunneum on V. longisporum Plant Colonization and Disease Development

*Metarhizium brunneum* inhibited in vitro colony growth of *V. longisporum* when grown in dual culture. The inhibition was already evident after 14 d, when the fungal pathogen area was 13% smaller in dual culture than in single culture. By day 23, the growth inhibition was 42% (Appendix A, Figure 1a). Vice versa, *V. longisporum* did not affect *M. brunneum* growth when grown in dual culture. The colony area of *M. brunneum* was marginally higher at 17 d in the dual culture (*p* = 0.091; F_1,8_ = 3.68; Appendix A). Increased mycelia production (17 d), and localized spore formation (23 d) were observed on the side confronting the *V. longisporum* colony (Figure 1b). 

### 3.2. Effect of Co-Inoculation M. brunneum on V. longisporum Disease Development

Co-inoculation with *M. brunneum* reduced the severity of symptoms of *Verticillium* stem striping disease. Although the shoot biomass of plants co-inoculated with both fungi was lower than that of control plants at 28 dpi, plants inoculated with both fungi had higher root dry weight (Figure 2b) and larger leaf area (Figure 2c) than plants inoculated with *V. longisporum* alone. The lower disease severity in plants co-inoculated with both fungi was reflected in the disease score which was lower at 21 and 28 dpi (Figure 2d). Furthermore, there was a delay in the colonization of the hypocotyl by the fungal pathogen in plants co-inoculated with *M. brunneum*. Specifically, the concentration of *V. longisporum* DNA increased between 21 and 28 dpi in plants inoculated with the fungal pathogen alone, while plants co-inoculated with both fungi had a lower amount of *V. longisporum* DNA at 28 dpi (Student’s *t*-test, *t* = −2.35, *df* = 7.24, *p* = 0.025).

However, the protective effects of *M. brunneum* co-inoculation were no longer observed at 35 dpi, where there were no significant differences in plant growth parameters, disease score or pathogen DNA in the hypocotyl between plants inoculated with *V. longisporum* alone and those co-inoculated with both fungi. We also measured *M. brunneum* in the hypocotyl tissues, which was detected only at 21 dpi (Figure 2f), with more plants showing endophytic growth in the dual inoculation treatment (three out of five plants) than in the treatment with only *M. brunneum* (one out of five).

### 3.3. Direct and Systemic Effects of M. brunneum on V. longisporum Plant Colonization and Disease Development

The split-root experiment revealed both local and systemic effects of *M. brunneum* inoculation on *V. longisporum* DNA levels in the roots and hypocotyl, as well as on plant biomass and disease progression.

Both fungi were detected only in the part of the root where they were inoculated. The biomass of *M. brunneum* in the rhizosphere, measured as the number of colony forming units (CFUs), was not affected by presence of *V. longisporum*. However, *M. brunneum* CFUs changed over time, decreasing from 7 to 21 dpi and increasing at 35 dpi (Figure 3a). The amount of *M. brunneum* DNA in the roots was found to be significantly higher in the local compartment where *V. longisporum* was present, compared to roots where the entomopathogen was alone or in the systemic compartment at 7 dpi (Figure 3b).

DNA of *M. brunneum* in the hypocotyl was low and detected in few plants; it was detected in one plant per treatment at seven and 21 dpi and in about 30% of the plants at 35 dpi. There were no significant differences between treatments, but there was a tendency for a higher amount of DNA in the local treatment at 35 dpi (Figure 3c).

*Verticillium longisporum* concentration in plant roots was highly variable during the first three weeks (7 and 21 dpi) and was not affected by the presence of *M. brunneum.* However, pathogen DNA was lower in the local compartment at 35 dpi (Figure 3e, *p* = 0.037) and lower, though not significantly, in the systemic compartment (*p* = 0.072). Pathogen DNA in the hypocotyl (Figure 3f) was detected at 7 dpi, but only in one (Local) or two (Systemic) plants. At 35 dpi, there was a lower amount of *V. longisporum* DNA in the local and systemic treatments, but differences were only marginally statistically significant (*p* = 0.061 and *p* = 0.092, respectively).

Inoculation with *M. brunneum* reduced plant stunting caused by *V. longisporum* infection as well as the AUDPC in both local and systemic treatments (Figure 4). We only observed effects on plant biomass at 35 dpi, where the *V. longisporum* treatment had lower biomass than the other treatments (Figure 4a, lm, F_4,50_ = 3.28, *p* = 0.018; Vl vs. Control *p* = 0.006). This was also observed for disease progression, where the *V. longisporum* treatment had a higher AUDPC (Figure 4b, lm, F_4,50_ = 3.62, *p* = 0.002).

### 3.4. Direct and Systemic Plant Responses to V. longisporum and M. brunneum: Gene Expression

Inoculation with *M. brunneum* alone affected the plant’s SA pathway by upregulating the SA downstream signaling gene *PR1*, but not the SA biosynthesis genes *PAL* and *ICS*, compared to the control. Likewise, the JA/ET downstream signaling gene *PDF1.2* was upregulated in the local and systemic root compartments (Figure 5d). *Metarhizium brunneum* inoculation also induced changes in the ET pathway, upregulating ET synthesis (*ACO*) in the local compartment (Figure 5e) and decreasing the downstream signaling gene *ERF* both in the local and systemic compartment. The plant responded to *V. longisporum* infection by induction of the SA-responsive gene *PR1* (Figure 5c). The JA/ET downstream marker *PDF1.2* was also induced in the roots in both local (Vl-L) and systemic (Vl-S) compartments (Figure 5d). Interestingly, there was only a systemic but not a local plant response to the pathogen in the ET biosynthesis pathway with regard to *ACO* induction (Figure 5e), while *ERF* was slightly downregulated when compared with control roots (Figure 5f). There were no changes in the expression of genes involved in the biosynthesis of JA (*AOS*, Appendix A), abscisic acid (*ABA2*, Figure 5g) or glucosinolates (GSL) (*CYP83A1* and *CYP79B2*, Figure 5h,i).

Interestingly, when *M. brunneum* was applied together in the same root compartment as *V. longisporum* (Mb-L/VI-L), roots had significantly higher *PAL* expression when compared with the control (Figure 5a). The induction of *PR1* in *V. longisporum* infected roots increased when *M. brunneum* was present in the systemic compartment (Vl-L/Mb-S), compared to the treatment with the pathogen alone (VL-L, Figure 5c). In the local treatment (Vl-L/Mb-L), induction of *PR1* was intermediate between the pathogen-only treatment and the systemic treatment and did not significantly differ from them. There was no local or systemic inhibition of the *ERF* gene (Figure 5f) when both fungi were colonizing the plant. The abscisic acid biosynthesis gene *ABA2* was induced in the compartments where *M. brunneum* was present (Mb-L/Vl-L and Mb-L/Vl-S, Figure 5g). In the systemic treatment with *M. brunneum* and *V. longisporum* in separate root compartments, significant downregulation of the aliphatic GSL biosynthesis gene *CYP83A1* was observed in the Vl compartment (VI-L/Mb-S, Figure 5h), while for the indol GSL biosynthesis gene *CYP79B2* downregulation was found in the Mb compartment (VI-L/Mb-S, Figure 5i).

## 4. Discussion

This study examined how *M. brunneum*, a soil-borne fungal entomopathogen, protects oilseed rape plants from *V. longisporum*, a soil-borne pathogen. The results show that *M. brunneum* effectively delays the colonization of plant roots by *V. longisporum*, leading to a significant reduction in disease symptoms. The research highlights the importance of direct competition and antibiosis in this process, as *M. brunneum* thrives in the presence of the pathogen while inhibiting its growth. Additionally, using a split-root setup, we observed that *M. brunneum* has a notable impact on the root’s defense response both locally and systemically by priming the SA pathway.

The in vitro confrontation assays conducted in this study demonstrated that *M. brunneum* inhibited the growth of *V. longisporum* by forming an inhibition halo. This inhibition may be attributed to the production of secondary metabolites with antibiotic effects [27]. Fungal inhibition in dual cultures with *Metarhizium* species was observed against a variety of other plant pathogens [20,21,22,54]. Crude extracts and partially purified fractions of *Metarhizium* have also been reported to inhibit fungal pathogens in previous studies [20,54,55,56]. Moreover, *M. brunneum* volatile organic compounds (VOCs) also inhibit plant pathogenic fungi [57]. Interestingly, our results indicate that the presence of the pathogen *V. longisporum* did not hinder the growth of *M. brunneum* in the confrontation assay. In fact, there was a slight inclination towards increased colony growth. Moreover, in pot experiments, the presence of *V. longisporum* did not affect the growth of *M. brunneum* in the soil. On the contrary, the entomopathogen was more abundant in the roots when *V. longisporum* was present. These findings suggest that the presence of other fungal mycelia may stimulate the growth of *M. brunneum*, leading to increased colonization of the root and potentially slowing down the spread of the pathogen’s infection.

After penetrating the roots, *V. longisporum* colonizes the xylem vessels and grows into the shoot, where it stays restricted to single vessels until later, when the fungus starts its saprophytic phase by invading the stem parenchyma and producing microsclerotia in the shoot tissues [41,58]. In a previous study by Eynck et al. [41], it was found that *V. longisporum* is initially detected at low levels in *B. napus* plants within the first four weeks after inoculation. However, its presence sharply increases in the hypocotyl at 35 dpi, as determined by quantifying the amount of its DNA. We aimed to investigate the impact of *M. brunneum* on the colonization of *V. longisporum* in the root and hypocotyl. Our observations revealed that at the root level (measured only in the split-root setup), the abundance of the pathogen was significantly lower at 35 dpi in root compartments where *M. brunneum* was present. In addition, during the co-inoculation experiment, we noticed that the abundance of *V. longisporum* in the hypocotyl was lower at 28 dpi. These findings suggest that while there were no apparent signs of *V. longisporum* inhibition in the roots at 7 and 21 dpi, a competitive interaction in the root zone resulted in reduced colonization of both the roots and hypocotyls by the pathogen. As a consequence, we observed diminished disease symptoms and improved plant growth in both the co-inoculation and split-root experiments. These results indicate that the presence of *M. brunneum* influenced the colonization dynamics of *V. longisporum*. In a previous study, it was found that pre-inoculation with the endophyte *V. isaacii* Vt305 also led to reduced plant colonization and symptom development caused by *V. longisporum* [59]. Although the specific mechanism was not investigated, the authors proposed that induced resistance could be responsible for this observed effect.

Our results from the split-root experiment further support the potential involvement of induced resistance in addition to the direct control mechanisms discussed earlier. Interestingly, even when *M. brunneum* was inoculated in a separate compartment from the pathogen, a protective effect was observed in the systemic treatment. In this case, the AUDPC values were lower, and there was no significant decrease in plant biomass compared to the treatment with the pathogen alone. Additionally, there was a marginal decrease in the amount of pathogen DNA detected in both the roots and hypocotyls. 

Beneficial root-associated fungi can elicit induced systemic responses (ISR) in plants and prime them to respond more effectively against pathogens [4,31]. The impact of *Metarhizium* species on plant defenses is not consistent, as studies have shown contradictory results. For example, *M. anisopliae* M202-1 was found to suppress plant defenses in peanut roots [32], while different species of *Metarhizium* induced the SA and JA pathways in other plants [12,35,36]. In a study conducted previously, we observed no significant plant responses after five weeks of *M. brunneum* Gd12 inoculation [37]. However, our present study suggests that *M. brunneum* CB15III induces the upregulation of two key pathways: the SA pathway, indicated by an increased transcription of the *PR1* gene, and the JA branch, resulting in upregulation of *PDF1.2*. In contrast, we observed the downregulation of *ERF2*, which is a protein involved in the transcriptional regulation of defense genes in response to ET and/or elicitors [60]. The inhibition of ET response has been observed both in *M. anisopliae* and the pathogen *F. oxysporum* [32]. This downregulation could potentially be a common response to root colonizers in plants.

ABA, SA and JA/ET pathways play a crucial role in the response of oilseed rape plants to *V. longisporum*, as demonstrated by comparative transcriptomic analysis [48]. Infection by *V. longisporum* results in the suppression of genes involved in ABA biosynthesis, induction of the SA signaling pathway and decreased response of the JA/ET pathway.

Another study investigated the phenylpropanoid and SA pathways in a resistant and susceptible variety of *B. napus*, demonstrating that the resistant line exhibited increased SA levels and elevated expression of SA marker genes *PR1* and *PR2* [49]. SA signaling activation was further supported by a study showing a two-fold increase in SA-activated *PR1* expression in *A. thaliana*. Our findings align with these results, as we observed higher *PR1* expression in both the local and systemic compartments of *V. longisporum*-inoculated plants. Furthermore, this study provides additional evidence supporting the role of the JA/ET signaling pathway in the plant response to *V. longisporum*. We found upregulation of the JA/ET downstream signaling gene *PDF1.2* at both the local and systemic levels. A previous study reported the activation of *PDF1.2* in oilseed rape [61] and *A. thaliana* but not in *B. napus* plants at 6 dpi [48]. Interestingly, we observed changes in ET markers exclusively in the systemic compartment. While the induction of *ACO* suggests activation of ET synthesis, the downregulation of *ERF* indicates inhibition of downstream signaling. Behrens et al. [48] also reported the induction of ethylene receptor 2 (*ETR2*) in *B. napus* at 3 dpi and *A. thaliana* at 6 dpi. The distinct systemic response of *PDF1.2*, *ACO* and *ERF* to *V. longisporum* inoculation suggests that the pathogen may locally manipulate JA/ET signaling while the plant is capable of activating systemic signaling.

Root associated beneficial fungi can modify the plant response to *V. longisporum*. For example, *Trichoderma harzianum* and *Bacillus velezenis* induced a priming response on *B. napus* against this pathogen that involved the activation of the JA and ET hormonal pathways [61]. Our observations suggest that *M. brunneum* altered the root response to *V. longisporum* both locally and systemically. Phenylalanine ammonia-lyase (PAL) is the first enzyme in the phenylpropanoid pathway, which is directly linked to increased resistance to *V. longisporum* [49]. We found that *M. brunneum* co-inoculation with the pathogen increased *PAL* gene expression. This is consistent with other studies showing that beneficial microorganisms can induce *PAL* gene expression or enzyme activity in response to a pathogen. For example, *P. fluorescens*, different strains of *B. bassiana* and *T. asperellum* T-203 induced *PAL* expression or activity in olive, tomato and cucumber plants infected with *V. dahliae*, *Rhizoctonia solani* and *Pseudomona syringae*, respectively [62,63,64].

Furthermore, we found that the gene expression of *PR1* was higher in *Verticillium*-infected roots when the entomopathogenic fungus was present in the adjacent compartment. This suggests that the plant mounts a stronger response to the pathogen through the SA pathway, which leads to a systemic induction of plant defenses. Although priming of plants against pathogens through the SA pathway has been reported for other beneficial endophytes, our study is the first to report priming in response to *Metarhizium* inoculation.

It is known that during the early stages of infection, *V. longisporum* requires the suppression of ABA biosynthesis to establish itself [48]. However, in our study we did not observe any downregulation of *ABA2* by *V. longisporum*. Instead, we found that *M. brunneum* induced an upregulation of this gene both locally and systemically when the pathogen was present. This finding suggests that the presence of *M. brunneum* may interfere with the pathogen’s manipulation of ABA biosynthesis, which could lead to increased resistance in the plant.

Glucosinolates (GSL) are secondary metabolites found in *Brassica* plants that serve as a defense mechanism [65] by exhibiting antimicrobial properties [66,67]. These compounds appear to be involved in the plant’s response to *V. longisporum*. For instance, the gene *CYP79B2*, that encodes for an enzyme involved in indolic GSL biosynthesis, is induced in *A. thaliana* plants when infected with the pathogen. Furthermore, the double mutant *cyp79b2 cyp79b3* showed greater susceptibility to this pathogen [68,69]. GSLs also play a role in endophyte-plant interactions, as the root endophyte *Piriformospora indica* DSM 11827 induces the *CYP79B2* gene. In its absence, as found in mutant *cyp79B2 cyp79B3*, the endophyte grows uncontrollably [70]. In our study, we found no evidence of *CYP79B2* or *CYP83A1* induction by *V. longisporum* in *B. napus* plants. Instead, we observed inhibition of *CYP83A1* in the compartment inoculated with the pathogen when *M. brunneum* was present in the adjacent compartment and inhibition of *CYP79B2* in the compartment inoculated with *M. brunneum*. These results suggest a possible synergistic inhibition of GSL biosynthesis by the two microorganisms. However, further investigations are needed to confirm this hypothesis.

Beneficial fungi can also induce plant resistance through the release of various metabolites in the zone of interaction [31,71]. These metabolites can be recognized as microbe-associated molecular patterns (MAMPs) by the plant immune system and trigger defense responses [31,72]. The mechanisms behind plant recognition of endophytic entomopathogenic fungi (EEF) and the induction of systemic resistance are not yet fully understood [7], but recent research suggests that the EEF *B. bassiana* (BG11 and FRh2) may elicit the upregulation of pattern recognition receptors in *A. thaliana* plants, indicating a potential for microbe-associated molecular pattern-triggered immunity [7,33]. There is growing evidence that EEF induces systemic resistance in the plant, both against insects and plant pathogens [33,36,37,38]. An important step in the future is to understand how these defense responses are elicited and to find MAMPs or other compounds responsible for the induction of plant defenses.

## 5. Conclusions

This study shows that *M. brunneum* delayed *V. longisporum* oilseed rape root colonization, resulting in decreased disease symptoms. The mechanisms involved include a faster colonization of *M. brunneum*, whose growth was stimulated by the pathogen’s presence. This has likely resulted in preempting the space and a competition for resources, together with antibiosis. Moreover, *M. brunneum* changed the plant’s response to the pathogen by locally activating *PAL* and *ABA* genes, suggesting an activation of the phenylpropanoid and abscisic acid pathways. Furthermore, with the split-root experiment we were able to prove that there is a plant-mediated effect, seen by improved plant growth and decreased disease symptoms when *M. brunneum* was in the systemic compartment. In addition, enhanced systemic induction of *PR1* suggested a priming effect. So far, several studies have demonstrated the induction of systemic resistance by EEF. To our knowledge, we provide the first evidence of ISR by *M. brunneum* against a soil-borne pathogen.

## Figures and Tables

**Figure 1 jof-09-00796-f001:**
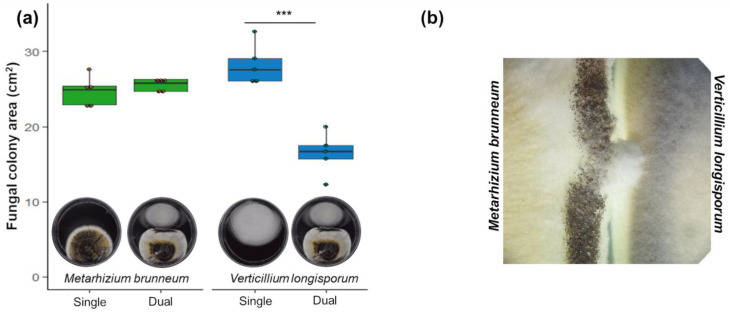
In vitro inhibition of *Verticillium longisporum* growth by *Metarhizium brunneum*. (**a**). Fungal colony area in single vs dual-culture plates; (**b**) detail of the confrontation zone. Fungal colonies grew in potato dextrose agar (PDA) media for 23 days either alone (single) or in confrontation with each other (dual). F_1,8_ = 44.8; *** *p* < 0.001, *n* = 4. (**b**) Close-up of the confrontation zone.

**Figure 2 jof-09-00796-f002:**
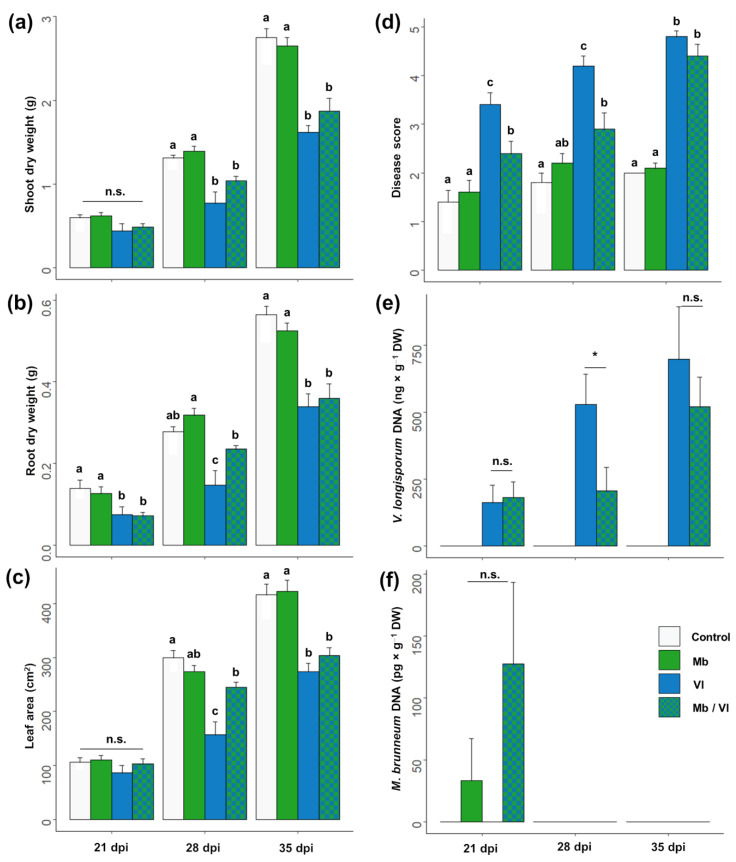
Effect of co-inoculation of oilseed rape roots with *M. brunneum* and *V. longisporum* on plant development and Verticillium disease progress. (**a**) Plant shoot dry weight; (**b**) Plant root dry weight; (**c**) Leaf area; (**d**) Disease score (see [41]; (**e**) *V. longisporum* DNA in hypocotyl; (**f**) *M. brunneum* DNA in hypocotyl. Plants were inoculated by root dipping 21 days after germination. For the dual inoculation (Mb/Vl), a mix of spores was used. Plants were harvested at 21, 28 and 35 days post-inoculation (dpi). Letters represent statistically significant differences within each date according to a linear model and Tukey’s post hoc test. * *p* < 0.05, Welch two sample *t*-test; n.s. = no significant differences. Bars represent means ± SE; *n* = 5.

**Figure 3 jof-09-00796-f003:**
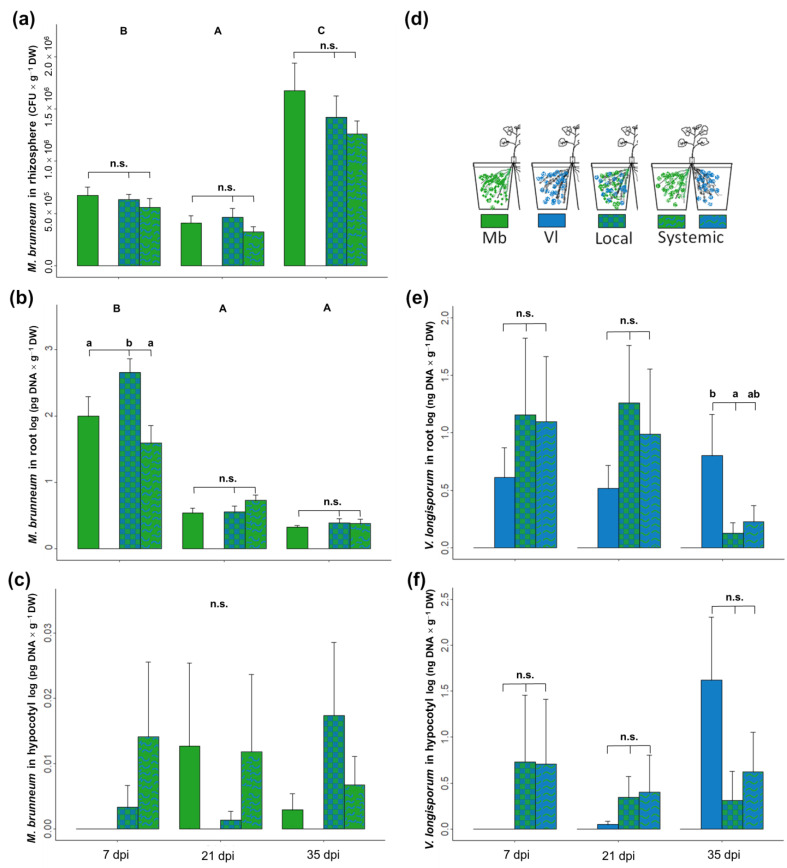
Colonization of rhizospheric soil (**a**), oilseed rape roots (**b**,**e**) and hypocotyls (**c**,**f**) by *M. brunneum* and/or *V. longisporum*. Plants grew in a split-root setup and were inoculated with fungal spores by root drenching. Treatment commenced with *M. brunneum* at transplanting (8 d) and was followed by *V. longisporum* after 7 d. Plants were harvested at 7, 21 and 35 dpi of *V. longisporum*. Panel (**d**) shows inoculation scheme and color patterns. Capital letters denote statistical differences between sampling dates; small letters denote significant differences among treatments within the same date. HDS Tukey; *p* < 0.05 according to a linear model; n.s. = no significant differences; bars represent means ± SE; *n* = 8.

**Figure 4 jof-09-00796-f004:**
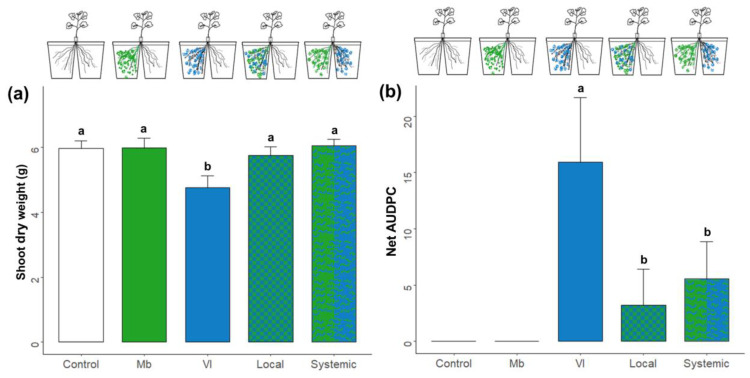
Shoot biomass (**a**) and area under the disease progress curve (AUDPC) (**b**) of oilseed rape plants inoculated with *V. longisporum* and/or *M. brunneum.* Plants grew in a split-root setup and were inoculated with fungal spores by root drenching. Treatment commenced with *M. brunneum* at transplanting (8 d) and was followed by *V. longisporum* after 7 d. Plants were harvested 35 d post-inoculation (dpi) of *V. longisporum*. Inoculation scheme of a given treatment is represented above each bar. Letters represent statistically significant differences within treatments according to a linear model and Tukey’s post hoc test. Bars represent means ± SE; *n* = 8.

**Figure 5 jof-09-00796-f005:**
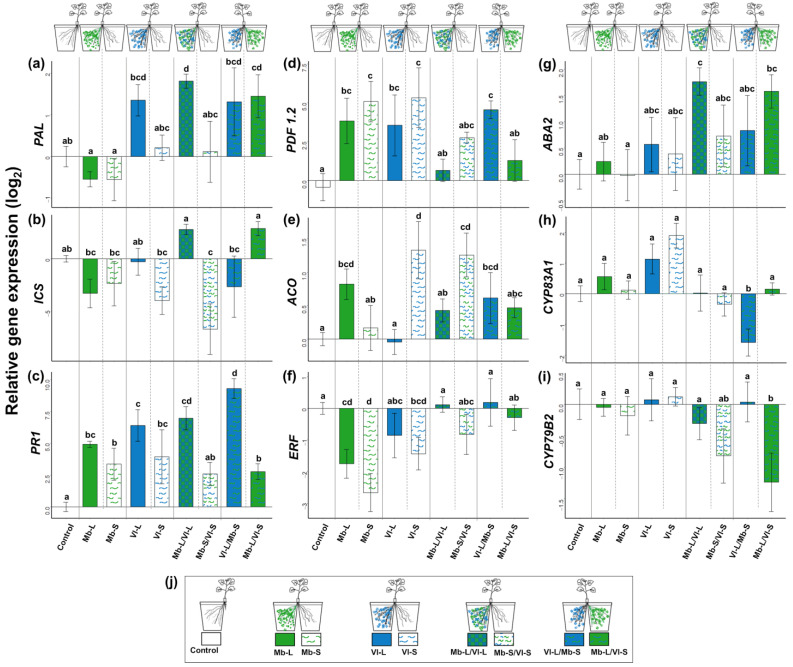
Expression of genes involved in defense signaling in roots of *B. napus* in response to *V. longisporum* and/or *M. brunneum* inoculation in a split-root setup. The left panel shows marker genes in the salicylic acid pathway: (**a**) *PAL*, phenylalanine ammonia lyase; (**b**) *ICS*, isochorismate synthase and (**c**) *PR1*, pathogenesis-related protein 1. The middle panel shows (**d**) marker gene in the jasmonic acid pathway *PDF1.2* plant defensin 1.2, and for ethylene biosynthesis (**e**) *ACO*, 1-Aminocyclopropane-1-Carboxylic Acid Oxidase and downstream signaling (**f**) *ERF2*, ethylene response factor 2. The right panel displays genes involved in (**g**) abscisic acid biosynthesis *ABA2*, xanthoxin dehydrogenase; (**h**) aliphatic glucosinolate (GSL) *CYP83A1*, cytochrome P450 83A1 and (**i**) indol-GSL biosynthesis *CYP79B2*, cytochrome P450 79B2. The illustrations above the figures a, d and g show the fungal inoculation scheme, either with *M. brunneum* (Mb) or *V. longisporum* (Vl) in the local (-L) or adjacent (-S) compartment and either treatments in the same compartment (Mb-L/Vl-L) or in each adjacent compartment of the same plant (Vl-L/Mb-S; Mb-L/Vl-S). Fungi were inoculated by root dipping. Treatment commenced with *M. brunneum* at transplanting (8 d) and was followed by *V. longisporum* after 7 d. Plants were harvested 7 days after *V. longisporum* inoculation. Panel (**j**) shows inoculation scheme and color patterns. Different letters indicate significant differences among treatments (Fisher LSD, *p* < 0.05). Bars represent means ± SE; *n* = 4–7.

## Data Availability

Data will be available upon request.

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
