# Peer review of "Local Competition and Enhanced Defense: How Metarhizium brunneum Inhibits Verticillium longisporum in Oilseed Rape Plants"

_jof, 2023, doi:10.3390/jof9080796_

Round 1
Reviewer 1 Report
Line306 The confrontation between Verticillium longisporum and Metarhizium brunneum was measured on a PDA tablet. As far as I know, the complete medium is rich in trace elements and amino acids. For example, iron ion is an important trace element affecting the growth and development of pathogenic fungi (Peng et al., 2022 Aug; 24(8):3693-3704. doi: 10.1111/1462-2920.16032.Epub 2022 May 6.PMID: 35523457). It is recommended to supplement the experiment of the two fungi on several other media, and discuss the other conditions that affect the inhibition of V. longisporum by M. brunneum on the plant.
Line377 The statistical significance markers for Figures 4A and B are suggested with the letters a,b, and c.
Line433 The experimental design of this study is more ingenious, especially the split-root experiment . But at the same time, it also faces a variety of comparison and further analysis. For example, this study involves multiple sets of interactions, such as, Mb-inhibitor-Vl, Mb-promoter-Plant, Mb-promoter-Plant-inhibitor-Vl, Vl-inhibitor-Plant, Vl-activation-Mb, Plant-Defense-VL, etc. However, for these three interactions (Mb-inhibitor-Vl, Mb-promoter-Plant, Mb-promoter-plant-inhibitor-Vl), there is no clearer explanation and distinction in this paper. Or draw on interactions between other pathogens and plants.
Reviewer 2 Report
The paper is devoted to the inhibition of the phytopathogenic fungus by Metarhizium brunneumm by both direct antagonism and through induced plant resistance. I read this work with interest and pleasure. The adequacy of the description of the methods and results is not in doubt. English is completely understandable. The conclusions are consistent with the results.
I have only minor points for this work
1. Indicate the Latin name and family of the plant in the abstract and keywords
2. L119-121. Were M. brunneum and B. longisporum detected in the non-sterile substrates used for plant cultivation? For example, was a CFU analysis done?
3. L209-216. Was the soil weighed for CFU analysis?
4. L232. Please specify if these primers are specific for Metarhizium brunneum or for the genus Metarizium, or for the PARB clade, etc.
5. In Figure 5, increase the font on the y-axis. Also use black (not grey) for the y-axis line and values
6. L 443-447. Antagonistic effect can also be due to volatile compounds. Please see recent work on M. brunneum DOI: 10.3390/jof8040326
